# The Effect of α-Fe_2_O_3_(0001) Surface Containing Hydroxyl Radicals and Ozone on the Formation Mechanism of Environmentally Persistent Free Radicals

**DOI:** 10.3390/toxics12080582

**Published:** 2024-08-10

**Authors:** Danli Liang, Jiarong Liu, Chunlin Wang, Kaipeng Tu, Li Wang, Lili Qiu, Xiuhui Zhang, Ling Liu

**Affiliations:** 1Key Laboratory of Cluster Science, Ministry of Education of China, School of Chemistry and Chemical Engineering, Beijing Institute of Technology, Beijing 100081, China; liangdanli111@126.com (D.L.); zhangxiuhui@bit.edu.cn (X.Z.); 2Key Laboratory of National Land Space Planning and Disaster Emergency Management of Inner Mongolia, School of Resources, Environment and Architectural Engineering, Chifeng University, Chifeng 024000, China; 3State Key Laboratory for Structural Chemistry of Unstable and Stable Species, CAS Research/Education Center for Excellence in Molecular Sciences, Institute of Chemistry, Chinese Academy of Sciences, Beijing 100190, China; 4Norinco Group Shanxi North Xingan Chemical Industry Company Limited, Taiyuan 030008, China

**Keywords:** environmentally persistent free radicals (EPFRs), α-Fe_2_O_3_(0001), ·OH, O_3_

## Abstract

The formation of environmentally persistent free radicals (EPFRs) is mediated by the particulate matter's surface, especially transition metal oxide surfaces. In the context of current atmospheric complex pollution, various atmospheric components, such as key atmospheric oxidants ·OH and O_3_, are often absorbed on particulate matter surfaces, forming particulate matter surfaces containing ·OH and O_3_. This, in turn, influences EPFRs formation. Here, density functional theory (DFT) calculations were used to explore the formation mechanism of EPFRs by C_6_H_5_OH on α-Fe_2_O_3_(0001) surface containing the ·OH and O_3_, and compare it with that on clean surface. The results show that, compared to EPFRs formation with an energy barrier on a clean surface, EPFRs can be rapidly formed through a barrierless process on these surfaces. Moreover, during the hydrogen abstraction mechanism leading to EPFRs formation, the hydrogen acceptor shifts from a surface O atom on a clean surface to an O atom of ·OH or O₃ on these surfaces. However, the detailed hydrogen abstraction process differs on surfaces containing oxidants: on surfaces containing ·OH, it occurs directly through a one-step mechanism, while, on surfaces containing O_3_, it occurs through a two-step mechanism. But, in both types of surfaces, the essence of this promotional effect mainly lies in increasing the electron transfer amounts during the reaction process. This research provides new insights into EPFRs formation on particle surfaces within the context of atmospheric composite pollution.

## 1. Introduction

With rapid economic development and continuous industrialization, the issue of complex air pollution in the atmosphere has been gaining more attention. This complexity primarily arises from the coexistence of various air pollutants with high concentrations and their complex interactions. Among the various pollutants, environmental persistent free radicals (EPFRs) have garnered widespread attention from environmental researchers as a new type of pollutant [1,2]. They are commonly found in various media such as fly ash and fine particulate matter in the atmospheric environment [3,4,5] and even in the soil [6]. Unlike traditional free radicals, EPFRs are more persistent and stable, thereby posing a long-term potential hazard to the environment [1,7,8]. Their harm is primarily attributed to their strong ability to generate reactive oxygen species [4,9], which can cause damage to human DNA and thus lead to various diseases [10,11,12,13].

Based on the experimental results of electron paramagnetic resonance (EPR), EPFRs can be categorized into three types: phenoxy-type EPFRs, semiquinone-type EPFRs, and cyclopentadienyl-type EPFRs [1,2]. Among these, phenoxy-type EPFRs exhibit the greatest persistence and have the highest concentration in the atmospheric environment [1,2]. Generally, the formation process of phenoxy-type EPFRs are mediated through the interactions between the organic precursor phenol (C_6_H_5_OH) and the particulate surfaces in the atmosphere [14,15]. This process involves a redox reaction in which C_6_H_5_OH is oxidized to form phenoxy radicals [16]. Due to the process of EPFRs formation being mediated by the surfaces of atmospheric particulates, these surfaces have a significant impact on the formation mechanism, and the effects induced by different surfaces are also different. Previous theoretical studies have shown that EPFRs can be formed on a single atmospheric particulate surface, such as Fe_2_O_3_, Al_2_O_3_, and TiO_2_ surfaces [1]. However, a single type of atmospheric particulate cannot accurately reflect the real atmospheric environment. More importantly, under complex air pollution, the surface of an atmospheric particulate often does not have a clean and single composition, but rather adsorbs a large amount of the atmospheric constituents to form a particulate surface containing atmospheric components [17,18,19,20]. Then, these particles have different effects on the formation of EPFRs. For instance, compared to a clean soot surface, a soot surface with adsorbed hydroxyl radical (·OH) is more conducive to the formation of EPFRs [21]. And compared to a clean Al_2_O_3_ surface, this surface with adsorbed acidic and basic pollutants is more conducive to the formation of EPFRs [22]. In addition to the surfaces of soot and Al_2_O_3_, transition metal oxides, especially Fe_2_O_3_ [23], exist extensively and are found in high concentrations in atmospheric PM_2.5_ and fly ash from various combustion sources, which could effectively promote the formation of EPFRs [15]. Therefore, exploring the effect of transition metal oxide surfaces on the formation of EPFRs is necessary in the context of complex atmospheric pollution.

As is widely recognized, ·OH and ozone (O_3_) are prominent oxidants in the atmosphere [24]. They exhibit extremely high reactivity in chemical reactions, and thus play pivotal roles in complex air pollution. Furthermore, ·OH and O_3_ have been observed to quickly be adsorbed on the surface of transition metal oxides and some saline particles [25,26,27,28]. In particular, laboratory studies have found that ·OH adsorbed on the surface of transition metal oxides can promote the formation of EPFRs, indicating that the surface containing adsorbed atmospheric components might play a crucial role in the formation of EPFRs [14]. And recent field observation studies have found a prominent positive correlation between the concentrations of EPFRs and O_3_ [29], indicating that O_3_ might affect the formation of EPFRs. Currently, theoretical studies have shown that EPFRs can be formed by C_6_H_5_OH on single α-Fe_2_O_3_(0001) surfaces [23], but the effect mechanism of α-Fe_2_O_3_(0001) surfaces containing ·OH and O_3_ on the formation of EPFRs is not well understood. Therefore, to explore the effect of α-Fe_2_O_3_(0001) surfaces containing ·OH and O_3_ on the formation of EPFRs under conditions of complex atmospheric pollution, models with adsorbed ·OH and O_3_ on a α-Fe_2_O_3_(0001) surface are used to understand the underlying mechanism in the context of complex atmospheric pollution.

In this study, density functional theory (DFT) was used to investigate the effect of α-Fe_2_O_3_(0001) surfaces containing ·OH and O_3_ on the formation of EPFRs from precursor C_6_H_5_OH under conditions of complex atmospheric pollution. The aim of this study was to understand how α-Fe_2_O_3_(0001) surfaces containing ·OH and O_3_ affect the formation of EPFRs under combined atmospheric pollution, in order to provide a scientific basis and theoretical support for addressing the environmental and health issues caused by EPFRs.

## 2. Computational Details

All calculations using density functional theory (DFT) in this work were performed using the Vienna Ab-initio Simulation Package (VASP 5.4.4) code [30,31]. The generalized gradient approximation (GGA) in the form of the Perdew–Burke–Ernzerhof (PBE) functional was used to describe the exchange–correlation energy [32,33]. This was performed to accurately represent the periodic models’ properties. The interactions between valence electrons and ion cores were described using the projector augmented wave (PAW) method, with a plane wave basis set cutoff energy of 400 eV [34,35]. The Brillouin zone sampling was carried out using a *k*-point grid of (3 × 3 × 1). The convergence criteria for electronic self-consistent iterations and forces were set to 5 × 10^−6^ eV and 0.02 eV/Å, respectively. A smearing of 0.20 eV was used. To prevent interactions between periodic images, a vacuum layer thickness of 15 Å was applied in the z-direction. DFT-D3 dispersion corrections were employed to account for the van der Waals interactions between molecules. A Hubbard U term is included in the PBE functional (DFT + U), where the effective U value for the 3*d* orbitals of Fe is 4.6 eV [36]. To reveal the favorable reaction paths, the climbing image nudged elastic band (CI-NEB) method was used to calculate the reaction barrier, using eight images generated between the initial state (IS) and final state (FS) [37]. The convergence criteria for electronic self-consistent iterations and forces were also set to 5 × 10^−6^ eV and 0.02 eV/Å, respectively.

To evaluate the interactions between C_6_H_5_OH and α-Fe_2_O_3_(0001) surface, the adsorption energy (*E*_ads_) of the C_6_H_5_OH was calculated as follows:$$E_{\mathrm{ads}}=E_{\mathrm{total}}-E_{\mathrm{substrate}}-E_{\mathrm{adsorbate}}$$

For the clean α-Fe_2_O_3_(0001) surface, *E*_total_, *E*_substrate_, and *E*_adsorbate_ refer to the total energy of C_6_H_5_OH together with the α-Fe_2_O_3_(0001) surface, bare α-Fe_2_O_3_(0001) surface, and gas phase C_6_H_5_OH, respectively. When considering the co-adsorption of ·OH or O_3_ on the α-Fe_2_O_3_(0001) surface, *E*_total_, *E*_substrate_, and *E*_adsorbate_ refer to the total energy of C_6_H_5_OH and ·OH or O_3_ together with the α-Fe_2_O_3_(0001) surface, ·OH or O_3_ on the α-Fe_2_O_3_(0001) surface, and the gas phase C_6_H_5_OH, respectively. The more negative the *E*_ads_ value, the more exothermic the adsorption process is, indicating a stronger adsorption of C_6_H_5_OH on the α-Fe_2_O_3_(0001) surface.

Although α-Fe_2_O_3_ spherical particles exhibit various exposed surfaces, the (0001) surface was selected as the ideal surface model because it is the most stable among all the exposed α-Fe_2_O_3_ surfaces [38]. This surface has a (3 × 2) size, comprising three periodic layers. The bottom periodic layer was fixed and the top two periodic layers, as well as adsorbates, were relaxed. The surface was optimized before the species were adsorbed, and the corresponding surface morphology is shown in Appendix A.

## 3. Results

EPFRs formation through the reaction of precursor C_6_H_5_OH on the clean α-Fe_2_O_3_(0001) surface begins with its effective adsorption. Recently, a theoretical investigation calculated an adsorption energy of −68.5 kJ/mol for the precursor C_6_H_5_OH on this surface, indicating a strong adsorption [39] of C_6_H_5_OH on the clean α-Fe_2_O_3_(0001) surface. In addition, the studies also shown that the dissociation process of C_6_H_5_OH through the cleavage of its O-H bond is exothermic (reaction energy of −61.8 kJ/mol), and need to overcome a reaction barrier of 18.3 kJ/mol [23]. These results indicate that EPFRs formation through the dissociation of C_6_H_5_OH on the clean α-Fe_2_O_3_(0001) surface is both thermodynamically and kinetically favorable. However, it is important to note that a single α-Fe_2_O_3_(0001) surface might not accurately reflect the real atmospheric particulate environment, particularly in the context of the characteristics of current atmospheric complex pollution. Therefore, in the present study, the effect of α-Fe_2_O_3_(0001) surfaces containing the important atmospheric oxidants ·OH and O_3_ on the mechanism of EPFRs formation from C_6_H_5_OH was explored under the conditions of complex air pollution.

### 3.1. The Effect of α-Fe_2_O_3_(0001) Surface Containing ·OH and O_3_ on EPFRs Formation

#### 3.1.1. The Effect of α-Fe_2_O_3_(0001) Surface Containing ·OH on EPFRs Formation

Although experimental studies have demonstrated that a transition metal surface containing ·OH could promote EPFRs formation [15], the detailed formation process cannot be captured, and therefore, the formation mechanism is unclear. Hence, EPFRs formation on the α-Fe_2_O_3_(0001) surface containing ·OH has been studied using the CI-NEB method to reveal the structure and energy changes during the process, as shown in Figure 1 and Appendix A.

Figure 1 presents the energy profile of the reaction process and configuration of the key intermediates, and the other configurations in the reaction process are shown in Appendix A. It can be seen from Figure 1, the reaction of C_6_H_5_OH on the α-Fe_2_O_3_(0001) surface containing ·OH is exothermic, and the corresponding reaction energy is −14.6 kJ/mol, indicating that C_6_H_5_OH dissociation process is thermodynamically favorable. Furthermore, the relative energy decreases during the process from IS to FS, indicating that the dissociation of C_6_H_5_OH on this surface is barrierless and therefore spontaneous; that is, this dissociation process is kinetically favorable. During the dissociation reaction of C_6_H_5_OH, significant changes were observed in the bond length between the O atom and the H atom. From IS to FS, the length of O1-H1 increased from 1.115 Å to 1.698 Å, indicating the break of old bond; meanwhile, the length of O2-H1 was observed to shorten from 1.410 Å to 1.027 Å, indicating the formation of a new bond, and finally, forming the products EPFRs and H_2_O. Specifically, the surface of α-Fe_2_O_3_(0001) containing ·OH participates in the formation of EPFRs by abstraction H from C_6_H_5_OH through the surface OH.

#### 3.1.2. The Effect of α-Fe_2_O_3_(0001) Surface Containing O_3_ on EPFRs Formation

As one of the essential strong oxidants in the atmosphere, O_3_ has been shown to participate in various atmospheric chemical reactions, including heterogeneous chemical reactions on particle surfaces. And some explorations have found that it first adsorbs on some surfaces, such as NaCl and Al_2_O_3_ [25,27], and then participates in a multistep reaction, in heterogeneous atmospheric chemistry. Recently, field observation has found a significant positive correlation between the concentration of EPFRs and O_3_ in PM_2.5_, suggesting that O_3_ might play a role in promoting EPFRs formation from C_6_H_5_OH. However, the specific mechanism of this process remains unclear. Therefore, the effect of the α-Fe_2_O_3_(0001) surface containing O_3_ on EPFRs formation has been studied using the CI-NEB method, similar to the study of the surface containing ·OH.

Figure 2 illustrates the energy changes of the reaction process and the configuration of key intermediates, and the other configurations in the reaction process are shown in Appendix A. It can be seen from Figure 2 that the reaction of C_6_H_5_OH on the α-Fe_2_O_3_(0001) surface containing O_3_ is exothermic, and the corresponding reaction energy is −115.9 kJ/mol, indicating that this reaction is thermodynamically favorable. And the relative energy decreases during the process from IS to FS, demonstrating that the dissociation of C_6_H_5_OH on this surface is spontaneous and barrierless, making the dissociation process kinetically favorable. Simultaneously, in the reaction process, there were significant changes in the distance between the O and H atoms. From IS to FS, the bond length of O1-H1 and O2-H1 increased from 0.982 Å to 3.273 Å and from 2.928 Å to 0.977 Å, respectively, indicating the breaking of old bonds and the formation of new bonds, and finally, the forming of the products—EPFRs. Similar to the reaction process on the surface containing ·OH, the substances adsorbed on the surface (O_3_) replace the surface as a new receptor for H adsorption.

It is important to note that, when EPFRs are formed through the reaction of C_6_H_5_OH on the α-Fe_2_O_3_(0001) surface containing O_3_, O_3_ first dissociates into O_2_ and a surface O atom. Afterwards, the surface O atom effectively abstracts the H atom from C_6_H_5_OH, thereby facilitating the formation of EPFRs. This is different from the reaction mechanism of the surface containing ·OH, where ·OH directly extracts H from C_6_H_5_OH. The following result would further demonstrate the differences in mechanisms between the two surfaces. 

### 3.2. The Comparison of EPFRs Formation on the α-Fe_2_O_3_(0001) Surface Containing ·OH and O_3_

To understand the difference in EPFRs formation between the single α-Fe_2_O_3_(0001) surface and the α-Fe_2_O_3_(0001) surface containing ·OH and O_3_, and to further reveal the effect mechanism of ·OH and O_3_, our work has been compared with the previous investigation.

Firstly, it can be found that the reaction mechanism shifts from hydrogen abstraction by surface O atom to the more reactive O atom in ·OH and O_3_, and, consequently, the energy barrier decreases from 18.3 kJ/mol to barrierless. These findings suggested there was a promotional effect of the α-Fe_2_O_3_(0001) surface containing ·OH and O_3_ for EPFRs formation. Furthermore, to identify the nature of this promotional effect, the co-adsorption energies and Bader charge during the reaction process were analyzed.

As shown in Figure 3, the adsorption energy of C_6_H_5_OH on the clean α-Fe_2_O_3_(0001) surface was −68.5 kJ/mol [23]. When C_6_H_5_OH is adsorbed on the α-Fe_2_O_3_(0001) surface containing ·OH and O_3_, the adsorption energies of C_6_H_5_OH increase to −148.3 kJ/mol and −95.6 kJ/mol, respectively. This increased adsorption energy indicated that the interaction between C_6_H_5_OH and the surface containing ·OH and O_3_ is stronger, suggesting a higher likelihood of a reaction occurring.

During the formation of EPFRs, the nature of hydrogen abstraction is an inner sphere electron transfer. To gain a deeper understanding of the impact of surfaces containing ·OH and O_3_ during this process, the amount of electron transfer on these surfaces was analyzed and compared with that on clean surfaces. On α-Fe_2_O_3_(0001) surfaces containing ·OH or O_3_, the amounts of electron transfer in the formation of EPFRs, where O of ·OH and O_3_ abstracts H from C_6_H_5_OH, was 0.630 |e| and 1.182 |e| (as shown in Figure 4), respectively. In comparison, the amount of electron transfer during hydrogen abstraction by surface O atoms on clean α-Fe_2_O_3_(0001) surfaces was only 0.078 |e|. This significant increase in electron transfer indicates that the electron transfer process is more favorable on α-Fe_2_O_3_(0001) surfaces containing ·OH and O_3_, demonstrating the promotional effect of these surfaces.

Furthermore, it is noteworthy that the EPFRs formation process would significantly impact the electron transfer on the surface. As shown in Figure 5, the electron transfer amount on the surface also significantly increases during the reaction process, rising from 0.078 |e| on a clean surface to 0.741 |e| and 1.264 |e| on surfaces containing ·OH and O_3_, respectively. This is because, once ·OH and O_3_ adsorbed on the surface react with C_6_H_5_OH, it would affect not only the interaction between C_6_H_5_OH and the surface, but also the interaction between ·OH or O_3_ and the surface, thereby influencing the surface charge distribution. Since the adsorption of ·OH and O_3_ on the surface primarily occurs through their interaction with surface Fe atoms, this effect can be further verified by comparing the changes in the charge of the surface Fe atoms. As shown in Figure 6, compared to the electron transfer amount of 1.284 |e| on the α-Fe_2_O_3_(0001) surface Fe sites without other adsorbates, the presence of ·OH and O_3_ on the surface increases the electron transfer amounts of Fe sites to 1.646 |e| and 1.697 |e|, respectively. Hence, surfaces containing ·OH and O_3_ directly influence the interaction between C_6_H_5_OH and surfaces.

It is important to emphasize that although the surfaces containing ·OH or O_3_ both promote the formation of EPFRs, the mechanisms by which they do so are different, as revealed by the reaction pathways. As shown in Figure 1 and Figure 2, on surfaces containing ·OH, the reaction involves the direct abstraction of a H atom from C_6_H_5_OH by the adsorbed ·OH, which can be concluded as a single-step reaction mechanism. On surfaces containing O_3_, the reaction requires the decomposition of O_3_ into O atoms and then the abstraction of a H atom from C_6_H_5_OH, following a two-step reaction mechanism. This difference can be attributed to whether the ·OH or O_3_ have unpaired electrons. The ·OH is a radical molecule with an unpaired electron, making it inclined to directly abstract H atoms, even when adsorbed on the surface. In contrast, O_3_, without unpaired electrons, first reacts by decomposing on the surface to produce an O atom with lone pair electrons, and then abstracts a H atom.

Except for C_6_H_5_OH, a study has demonstrated that 2-monochlorophenol can undergo dissociation to form EPFRs on clean Fe_2_O_3_ surface, which was confirmed using EPR [15]. Considering the similarity in precursor structures, it is reasonable to assume that the 2-monochlorophenol may also dissociate to form EPFRs on α-Fe_2_O_3_(0001) surfaces containing ·OH and O_3_. Overall, compared to clean α-Fe_2_O_3_(0001) surfaces, the surfaces containing ·OH or O_3_ not only better represent the characteristics of particle surfaces in a real atmospheric environment but also reveal the promotional effect of these surfaces on the formation of EPFRs. Similar to the corresponding effect from clean α-Fe_2_O_3_ surfaces, it has also been observed that clean surfaces of other metal oxides, such as Al_2_O_3_ and TiO_2_, play a favorable role in EPFRs formation. And EPFRs formation on these surfaces follows the same conceptual mechanism, namely precursor adsorption followed by dissociation, as confirmed using EPR spectroscopy [1,2]. Hence, it is reasonable to expect that other similar metal oxide surfaces containing ·OH and O_3_ may also follow the same mechanism for EPFRs formation under conditions of complex atmospheric pollution, as indicated by parallel experiments using EPR spectroscopy.

In addition to the influence of the surfaces containing ·OH and O_3_, surfaces containing other atmospheric oxidants like O_2_ (the second most dominant component in the atmosphere) [40], may also affect EPFRs formation. Subsequently, the effect of α-Fe_2_O_3_ (0001) surface containing O_2_ on EPFRs formation has been studied (Appendix A). It can be seen from Appendix A that the reaction of C_6_H_5_OH on the α-Fe_2_O_3_ (0001) surface containing O_2_ is endothermic, with the corresponding reaction energy being 1.3 kJ/mol, indicating that this reaction is thermodynamically unfavorable. Therefore, in the atmospheric context, the promotional effect of a surface containing O_2_ on the formation of EPFRs may be infeasible.

Furthermore, surfaces may also adsorb species possessing the ability to transfer hydrogen atoms, such as water (H_2_O) and acidic and basic pollutants, thereby influencing the formation of EPFRs. The effect of the α-Fe_2_O_3_ (0001) surface containing H_2_O on EPFRs formation was further studied, as illustrated in Appendix A. The corresponding reaction energy and reaction barrier are −38.0 and 9.5 kJ/mol, respectively, indicating that EPFRs formation through the proton transfer mechanism is both thermodynamically and kinetically favorable. Given the similarity among species that possess the ability to transfer hydrogen atoms, it is reasonable to assume that the EPFRs could be formed on an α-Fe_2_O_3_ (0001) surface containing acidic and basic pollutants through the proton transfer mechanism. Additionally, EPFRs could be barrierless formed on the Al_2_O_3_ surfaces containing H_2_O, acidic pollutants, and basic pollutants through the proton transfer mechanism [22,41]. Considering the similarity in the mechanisms of EPFRs formation on various surfaces, these impacts might also occur on α-Fe_2_O_3_ (0001) surfaces. In this work, the α-Fe_2_O_3_ (0001) surfaces containing ·OH and O_3_ selected as the interface are beneficial to EPFRs formation through the hydrogen abstraction mechanism. This provides new insights into the formation of EPFRs in the context of atmospheric composite pollution.

Additionally, EPFRs could also exhibit certain reactivity and undergo transformation to form new species, such as toxic dioxins, which have a more negative impact on human health and the environment than EPFRs themselves. Therefore, long-term simulations are necessary and should be considered to reveal the evolution and stability of EPFRs over time in detail [42]. In the future, we will also conduct theoretical simulations to study the transformation process of EPFRs in the atmosphere and the influence of different atmospheric components on this transformation mechanism. This will help us gain a deeper understanding on the impact of EPFRs on the atmospheric environment and human health.

## 4. Conclusions

In this study, the effects of α-Fe_2_O_3_(0001) surfaces containing ·OH and O₃ on the formation mechanism of EPFRs from C₆H₅OH were investigated using DFT calculations. The results showed that, compared to a clean α-Fe_2_O_3_(0001) surface, these surfaces containing ·OH and O₃ could enhance EPFRs formation by reducing the energy barrier from 18.3 kJ/mol to barrierless. Meanwhile, the acceptor of the hydrogen abstraction mechanism shifts from the surface O atom to the O atom of ·OH or O_3_. 

Further analysis of the co-adsorption energy and Bader charge revealed that, compared to C_6_H_5_OH adsorbed on the clean α-Fe₂O₃(0001) surface, surfaces containing ·OH and O₃ not only enhance the stability of C_6_H_5_OH by increasing its adsorption energy, but also increase the electron transfer amounts during the reaction process, thereby promoting the hydrogen abstraction reaction of C_6_H_5_OH to form EPFRs. More importantly, the mechanisms by which surfaces containing ·OH and O_3_ influence the formation of EPFRs are different, as seen from the reaction pathways. On surfaces containing ·OH, the reaction proceeds via a single-step direct hydrogen abstraction mechanism, where ·OH abstracts a H atom from C₆H₅OH. In contrast, on surfaces containing O_3_, the reaction proceeds via a two-step mechanism, where O_3_ first decomposes into an O atom, which then abstract a H atom from C₆H₅OH. 

This research provides new insights into the formation mechanism of EPFRs on particle surfaces within the context of atmospheric composite pollution. Future research could further explore the impact of other environmental factors on EPFRs formation to gain a comprehensive understanding of formation mechanisms of atmospheric pollutants.

## Figures and Tables

**Figure 1 toxics-12-00582-f001:**
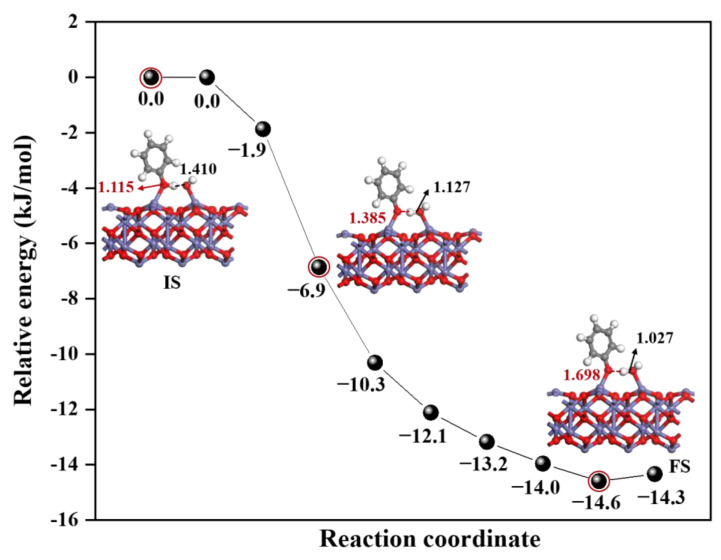
The energy profile for the reaction of C_6_H_5_OH on the α-Fe_2_O_3_(0001) surface containing ·OH together with the corresponding structures. Energy is in kJ/mol, and bond length is in Å. Color code: Fe (blue), C (gray), O (red), and H (white). Line: O1-H1 (red line), O2-H1 (black line).

**Figure 2 toxics-12-00582-f002:**
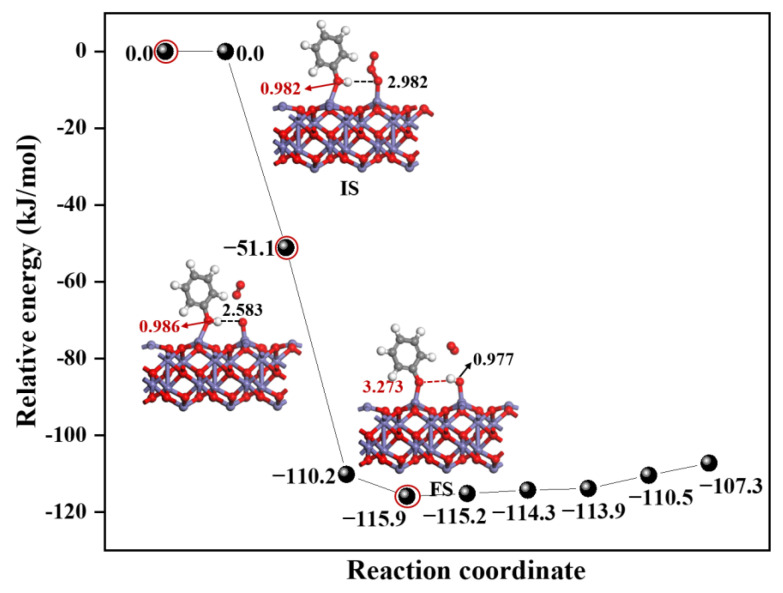
The energy profile for the reaction of C_6_H_5_OH on the α-Fe_2_O_3_(0001)) surface containing O_3_ together with the corresponding structures. Energy is in kJ/mol, and bond length is in Å. Color code: Fe (blue), C (gray), O (red), and H (white). Line: O1-H1 (red line), O2-H1 (black line).

**Figure 3 toxics-12-00582-f003:**
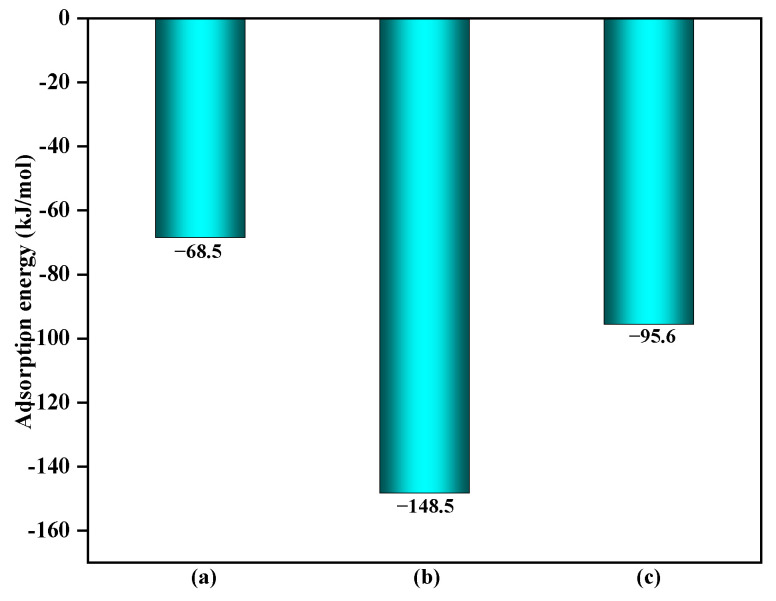
The adsorption energy of C_6_H_5_OH on the different surfaces: (**a**) clean α-Fe_2_O_3_(0001) surface, (**b**) α-Fe_2_O_3_(0001) surface containing ·OH and (**c**) α-Fe_2_O_3_(0001) surface containing O_3_. Energy is in kJ/mol.

**Figure 4 toxics-12-00582-f004:**
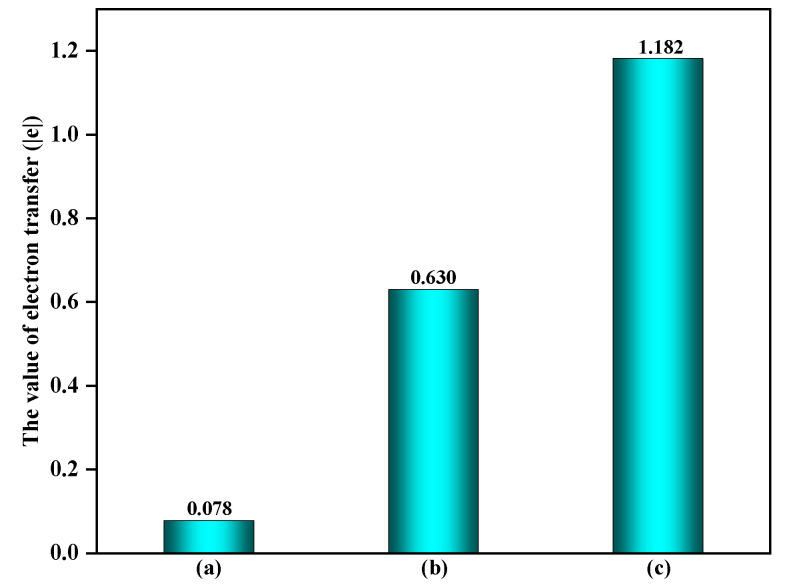
The amount of electron transfer during EPFRs formation: (**a**) clean α-Fe_2_O_3_(0001) surface, (**b**) α-Fe_2_O_3_(0001) surface containing ·OH, and (**c**) α-Fe_2_O_3_(0001) surface containing O_3_.

**Figure 5 toxics-12-00582-f005:**
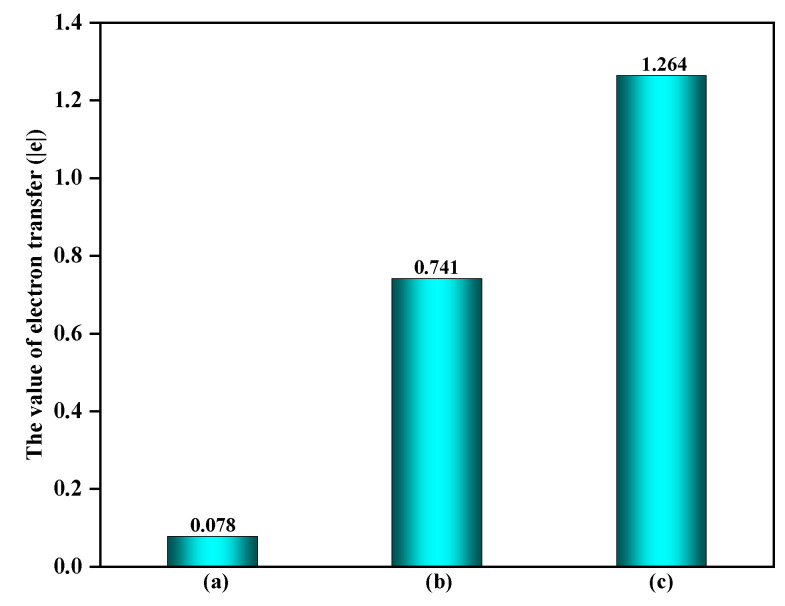
The amount of electron transfer of surface on the different surfaces: (**a**) clean α-Fe_2_O_3_(0001) surface, (**b**) α-Fe_2_O_3_(0001) surface containing ·OH, and (**c**) α-Fe_2_O_3_(0001) surface containing O_3_.

**Figure 6 toxics-12-00582-f006:**
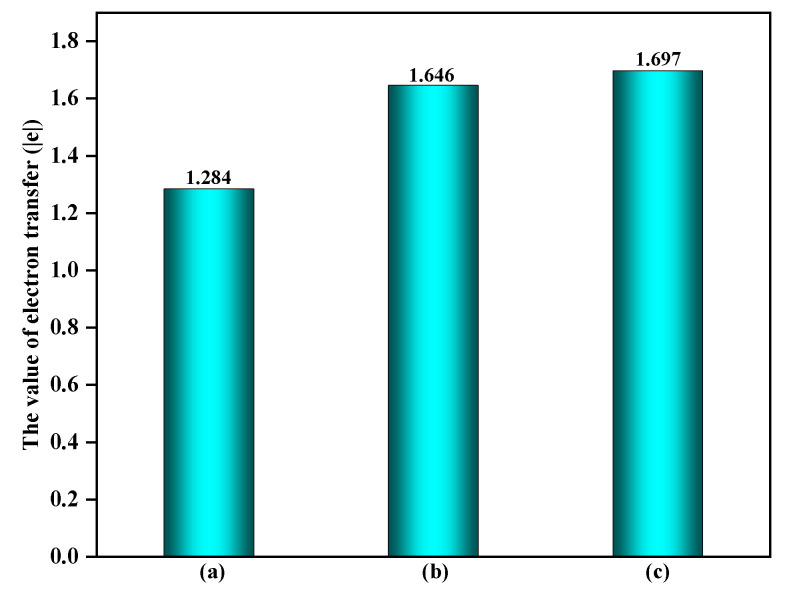
The amount of electron transfer of Fe site on the different surfaces: (**a**) clean α-Fe_2_O_3_(0001) surface, (**b**) α-Fe_2_O_3_(0001) surface containing ·OH, and (**c**) α-Fe_2_O_3_(0001) surface containing O_3_.

## Data Availability

The data presented in this study are available on request from the corresponding author.

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
