# Peer review of "The Effect of α-Fe2O3(0001) Surface Containing Hydroxyl Radicals and Ozone on the Formation Mechanism of Environmentally Persistent Free Radicals"

_toxics, 2024, doi:10.3390/toxics12080582_

Round 1

Reviewer 1 Report

Comments and Suggestions for Authors

The study investigates the mechanism of formation of environmentally persistent free radicals (EPFRs) on the α-Fe2O3(0001) surface containing hydroxyl radicals (·OH) and ozone (O3). Using density functional theory (DFT), researchers explored how these surfaces influence the formation of EPFRs from the precursor C6H5OH. The results show that the presence of ·OH and O3 on the α-Fe2O3(0001) surface reduces the energy barrier for EPFR formation, making the process barrierless. EPFR formation occurs through different mechanisms: ·OH facilitates direct hydrogen abstraction, while O3 first dissociates into oxygen atoms which then extract hydrogen. This study provides new insights into EPFR formation in the context of complex atmospheric pollution.

- Expanding the study to include other metal oxide surfaces commonly found in the atmosphere, such as Al2O3 and TiO2, could provide a more comprehensive understanding of the mechanisms of EPFR formation under different atmospheric conditions (Paragraphs "Introduction" and "Previous Studies").

- In order to improve the study consider referencing studies such as: "Modeling of an air quality monitoring network with high space-time resolution DOI 10.1016/B978-0-444-64235-6.50035-8"

- Integrating realistic environmental factors such as humidity, variable temperature, and the presence of other pollutants in the simulations could improve the practical relevance of the obtained results. This would help better reflect real atmospheric pollution conditions (Paragraphs "Computational Details" and "Discussion").

- Conducting parallel experiments to validate the theoretical results obtained from DFT simulations could enhance the robustness of the study. Experiments using techniques like electron paramagnetic resonance (EPR) spectroscopy could provide concrete evidence of the proposed mechanism (Paragraphs "Computational Details" and "Results").

- Investigating the effect of other atmospheric oxidants and volatile organic compounds (VOCs) on EPFR formation could offer a more complete view of the role of complex chemical interactions in the atmosphere (Paragraphs "Introduction" and "Results").

- Implementing long-term simulations to observe the evolution and stability of EPFRs over time, as well as their long-term effects on human health and the environment, could help in better understanding the long-term impact of EPFRs (Paragraphs "Discussion" and "Conclusions").

Author Response

Dear Editor and reviewers,

Thank you very much for your handling and comments on our manuscript “The Effect of α-Fe2O3(0001) Surface Containing Hydroxyl Radicals and Ozone on the Formation Mechanism of Environmentally Persistent Free Radicals” (Manuscript toxics-3114689). These comments are all valuable and very helpful for revising and improving our paper. We have revised the manuscript carefully according to your suggestions. The point-to-point responses are as following:   

Reviewer #1:

1) Expanding the study to include other metal oxide surfaces commonly found in the atmosphere, such as Al2O3 and TiO2, could provide a more comprehensive understanding of the mechanisms of EPFR formation under different atmospheric conditions (Paragraphs "Introduction" and "Previous Studies").

Response: Thanks for the reviewer’s valuable and professional comments. According to the reviewer’s suggestion, the discussion of other metal oxide surfaces has been added in the revised manuscript as follows:

  1. lines 55-56 of page 2 in the revised manuscript:

"Previous theoretical studies have shown that EPFRs can be formed on a single atmospheric particulates surface, such as Fe2O3, Al2O3 and TiO2 surfaces.".

  1. lines 279-286 of page 8 in the revised manuscript:

“Similar to the corresponding effect from clean α-Fe2O3 surface, it has also been observed that clean surfaces of other metal oxides, such as Al2O3 and TiO2, play a favourable role in EPFRs formation. And the EPFRs formation on these surfaces follow the same conceptual mechanism namely precursor adsorption followed by dissociation, as confirmed by EPR spectroscopy.1,2 Hence, it is reasonable to expect that other similar metal oxide surfaces containing ·OH and O3 may also follow the same mechanism for EPFRs formation under conditions of complex atmospheric pollution, as indicated by parallel experiments using EPR spectroscopy.”.

2) In order to improve the study consider referencing studies such as: "Modeling of an air quality monitoring network with high space-time resolution DOI 10.1016/B978-0-444-64235-6.50035-8"

Response: Thanks for the valuable and professional suggestion. The corresponding referencing studies have been added in lines 462-463 of page 12 in the revised manuscript as follows: " Sofia, D.; Giuliano, A.; Gioiella, F.; Barletta, D.; Poletto, M. Modeling of an Air Quality Monitoring Network with High Space-Time Resolution. In Computer Aided Chemical Engineering 2018, 43, 193–198.".

3) Integrating realistic environmental factors such as humidity, variable temperature, and the presence of other pollutants in the simulations could improve the practical relevance of the obtained results. This would help better reflect real atmospheric pollution conditions (Paragraphs "Computational Details" and "Discussion").

Response: Thanks for the valuable and professional suggestion. According to the reviewer’s suggestion, incorporating realistic environmental factors, such as humidity, variable temperature, and the presence of other pollutants, into the simulations could enhance the practical relevance of the results obtained. Thus, we further studied the effects of these environmental factors on EPFRs formation. Specifically, we firstly studied the effect of the α-Fe2O3 (0001) surface containing H2O on EPFRs formation. These results shown that the EPFRs could be formed on α-Fe2O3 (0001) surface containing H2O through the proton transfer mechanism. This result is similar to that on Al2O3 surface, where EPFRs could also be barrierless formed through the proton transfer mechanism.1 Except to the Al2O3 surfaces containing H2O, we note that there has been a study showing that EPFRs could be barrierless formed on the Al2O3 surfaces containing acidic and basic pollutants through the proton transfer mechanism.2 Additionally, another study has demonstrated that temperature also affects the EPFRs formation on the particle surface. With the reaction temperature increasing from 150 to 400 °C, the yields of EPFRs formed on the Fe2O3 surface by C6H5OH also increased. The highest yields were observed between 300 and 350 °C. These results were confirmed by EPR.3 Considering the similarity in the mechanisms of EPFRs formation on various surfaces, it is reasonable to assume that the impacts of environmental factors (humidity, variable temperature, and the presence of other pollutants) on the EPFRs formation on the α-Fe2O3 surfaces are similar to those observed on other surfaces. And the relevant comments have been added in lines 296-308 of page 8 in the revised manuscript as follows: “Furthermore, surfaces may also adsorb species possessing the ability to transfer hydrogen atoms, such as water (H2O), acidic and basic pollutants, thereby influencing the formation of EPFRs. The effect of α-Fe2O3 (0001) surface containing H2O on the EPFRs formation was further studied, as illustrated in Figure S5. The corresponding reaction energy and reaction barrier are -38.0 and 9.5 kJ/mol, respectively, indicating that the EPFRs formation through the proton transfer mechanism is both thermodynamically and kinetically favourable. Given the similarity among species that possess the ability to transfer hydrogen atoms, it is reasonable to assume that the EPFRs could be formed on α-Fe2O3 (0001) surface containing acidic and basic pollutants through the proton transfer mechanism. Additionally, EPFRs could be barrierless formed on the Al2O3 surfaces containing H2O, acidic pollutants and basic pollutants through the proton transfer mechanism.22,41 Considering the similarity in the mechanisms of EPFRs formation on various surfaces, these impacts might also occur on α-Fe2O3 (0001) surfaces.”.

Figure S5 The energy profile for the reaction of C6H5OH on the α-Fe2O3(0001) surface containing H2O together with the corresponding structures, including initial state (IS), transition state (TS), and final state (FS). Energy is in kJ/mol. Color code: Fe (blue), C (gray), O (red), and H (white).

4) Conducting parallel experiments to validate the theoretical results obtained from DFT simulations could enhance the robustness of the study. Experiments using techniques like electron paramagnetic resonance (EPR) spectroscopy could provide concrete evidence of the proposed mechanism (Paragraphs "Computational Details" and "Results").

Response: Thanks for the valuable and professional suggestion. According to the reviewer’s suggestion, the relevant comments have been added in lines 281-286 of page 8 in the revised manuscript as follows: “And the EPFRs formation on these surfaces follow the same conceptual mechanism namely precursor adsorption followed by dissociation, as confirmed by EPR spectroscopy. Hence, it is reasonable to expect that other similar metal oxide surfaces containing ·OH and O3 may also follow the same mechanism for EPFRs formation under conditions of complex atmospheric pollution, as indicated by parallel experiments using EPR spectroscopy.”. The experiments by electron paramagnetic resonance (EPR) spectroscopy should be considered to confirm the theoretical results by DFT simulations. However, since our research is conducted from a theoretical perspective and is limited by the software, it is not feasible to conduct parallel experiments. Our purpose of research lies in conducting theoretical prediction experiments. Nevertheless, the reviewer has provided us with invaluable insights for future work. Consequently, we plan to strengthen collaborations and partnerships with relevant experimental research groups both domestically and internationally, striving to address the questions raised by the reviewer.

5) Investigating the effect of other atmospheric oxidants and volatile organic compounds (VOCs) on EPFR formation could offer a more complete view of the role of complex chemical interactions in the atmosphere (Paragraphs "Introduction" and "Results").

Response: Thanks for the valuable and professional suggestion. According to the reviewer’s suggestion, we further studied the effect of α-Fe2O3 (0001) surface containing atmospheric oxidants O2 (the second most dominant component in the atmosphere) on the EPFRs formation.4 The corresponding results has been added in lines 287-295 of page 8 in the revised manuscript as follows: “In addition to the influence of the surfaces containing ·OH and O3, surfaces containing other atmospheric oxidants like O2 (the second most dominant component in the atmosphere),40 may also affect the EPFRs formation. Subsequently, the effect of α-Fe2O3 (0001) surface containing O2 on the EPFRs formation has been studied (Figure S4). It can be seen from the Figure S4, the reaction of C6H5OH on the α-Fe2O3 (0001) surface containing O2 is endothermic, with the corresponding reaction energy is 1.3 kJ/mol, indicating that this reaction is thermodynamically unfavourable. Therefore, in the atmospheric context, the promotion effect of surface containing O2 on the formation of EPFRs may be infeasible.” Except for C6H5OH, a previous study has shown that EPFRs can be formed by volatile organic compounds (VOCs) such as 2-monochlorophenol on Fe2O3, which was confirmed by EPR,3 suggesting similar formation on the surfaces containing ·OH and O3. The corresponding results has been added in lines 272-276 of page 8 in the revised manuscript as follows: “Except for C6H5OH, a study has demonstrated that 2-monochlorophenol can undergo dissociation to form EPFRs on clean Fe2O3 surface, which was confirmed by EPR.15 Considering the similarity in precursor structure, it is reasonable to assume that the 2-monochlorophenol may also dissociate to form EPFRs on α-Fe2O3(0001) surfaces containing ·OH and O3.”.

Figure S4 The energy profile for the reaction of C6H5OH on the α-Fe2O3(0001) surface containing O2 together with the corresponding structures. Energy is in kJ/mol. Color code: Fe (blue), C (gray), O (red), and H (white).

6) Implementing long-term simulations to observe the evolution and stability of EPFRs over time, as well as their long-term effects on human health and the environment, could help in better understanding the long-term impact of EPFRs (Paragraphs "Discussion" and "Conclusions").

Response: Thanks for the valuable and professional suggestion. We note that previous work has explored the evolution and stability of EPFRs. Generally, compared to common ·OH radicals, EPFRs often exhibit a certain degree of stability, while compared to other substances that do not contain single electron, EPFRs also exhibit certain reactivity. Based on the previous investigation, it could be found that EPFRs could undergo evolution through various pathways in the atmosphere. On the one hand, EPFRs can undergo polycondensation via self-quenching processes, leading to the formation of toxic dioxins, although this reaction occurs slowly and often requires more stringent conditions such as high-temperature environments. On the other hand, EPFRs can undergo transformation through reactions with coexisting components in the atmosphere at ambient temperatures. Studies have shown that once released into the atmosphere, EPFRs rapidly react with hydroxyl and nitrate radicals, leading to their evolution and ultimately affecting their stability in the atmosphere. Thus, the evolution of EPFRs over time would form new species, such as toxic dioxins, which have a negative impact on human health and the environment, and subsequently affect their stability. According to the reviewer’s suggestion, the relevant comments have been added in lines 312-317 of page 8 and lines 318-319 of page 8 in the revised manuscript as follows: "Additionally, EPFRs could also exhibit certain reactivity and undergo transformation to form new species, such as toxic dioxins, which have a more negative impact on human health and the environment than EPFRs itself. Therefore, long-term simulations are necessary and should be considered to reveal the evolution and stability of EPFRs over time detail.42 In the future, we will also conduct theoretical simulations to study the transformation process of EPFRs in the atmosphere and the influence of different atmospheric components on the transformation mechanism. This will help us gain a deeper understanding on the impact of EPFRs on the atmospheric environment and human health.".

References

(1)   Wang, W.; Zhang, R.; Liu, Z.; Wang, W.; Zhang, Q.; Wang, Q. Periodic DFT Calculation for the Formation of EPFRs from Phenol on γ-Al2O3 (110): Site-Dependent Mechanism and the Role of Ambient Water. Journal of Environmental Chemical Engineering 2022, 10 (5), 108386.

(2)   Wang, L.; Liang, D.; Liu, J.; Du, L.; Vejerano, E.; Zhang, X. Unexpected Catalytic Influence of Atmospheric Pollutants on the Formation of Environmentally Persistent Free Radicals. Chemosphere 2022, 303, 134854.

(3)   Vejerano, E.; Lomnicki, S.; Dellinger, B. Formation and Stabilization of Combustion-Generated Environmentally Persistent Free Radicals on an Fe(III)2O3 /Silica Surface. Environ. Sci. Technol. 2011, 45 (2), 589–594.

(4)   Pan, W.; Chang, J.; He, S.; Xue, Q.; Liu, X.; Fu, J.; Zhang, A. Major Influence of Hydroxyl and Nitrate Radicals on Air Pollution by Environmentally Persistent Free Radicals. Environ Chem Lett 2021, 19 (6), 4455–4461.

Reviewer 2 Report

Comments and Suggestions for Authors

The authors investigated the formation mechanism of EPFR under the conditions of air pollution and found that the formation of EPFR can be promoted by atmospheric radicals. I think this is significant as a case of applying the formation conditions of EPFR on the transition metal oxides in the particulate matters in the atmophere.

I have a few comments.

1) Air pollution particles have a spherical shape. Does this make a difference compared to the surface of FE bulk?

2) In Figures 1 and 2, is the barrier due to activation energy considered when going to iS and TS?

3) The particle surface may be covered with water in some places due to moisture in the atmosphere. Is there no difference in this case?

4) It might be helpful to briefly explain the purpose of each model the authors used.

Author Response

Dear Editor and reviewers,

Thank you very much for your handling and comments on our manuscript “The Effect of α-Fe2O3(0001) Surface Containing Hydroxyl Radicals and Ozone on the Formation Mechanism of Environmentally Persistent Free Radicals” (Manuscript toxics-3114689). These comments are all valuable and very helpful for revising and improving our paper. We have revised the manuscript carefully according to your suggestions. The point-to-point responses are as following:   

Reviewer #2:

1) Air pollution particles have a spherical shape. Does this make a difference compared to the surface of FE bulk?

Response: Thanks for the valuable and professional suggestion. Although spherical shape is a common morphological feature of atmospheric particles on a macro scale, spherical particles show different exposed surfaces. Among them, the (0001) facet is the most stable exposed surface of α-Fe2O3 particles. Thus, in this work, the (0001) surface was selected as the ideal surface model, because it is the most stable exposed surface among the various surfaces. The corresponding revision has been added in lines 122-124 of page 3 in the revised manuscript as follows: "Although α-Fe2O3 spherical particles exhibit various exposed surfaces, the (0001) surface was selected as the ideal surface model because it is the most stable among all the exposed surfaces α-Fe2O3.".

2) In Figures 1 and 2, is the barrier due to activation energy considered when going to IS and TS?

Response: Thanks for the valuable and professional suggestion. To calculate the reaction paths, the Climbing Image Nudged Elastic Band (CI-NEB) method was used, using eight images generated between the initial state (IS) and final state (FS). As shown in Figures 1 and 2, the energy profile of the reaction process shown that the relative energy decreases during the process from IS to FS, indicating that the dissociation of C6H5OH on this surface is barrierless process. Thus, the barrier is considered when going to IS and TS according to activation energy. The corresponding revision has been added in lines 107-109 of page 3 in the revised manuscript as follows: "To reveal the favorable reaction paths, the Climbing Image Nudged Elastic Band (CI-NEB) method was used to calculate the reaction barrier, using eight images generated between the initial state (IS) and final state (FS).".

3) The particle surface may be covered with water in some places due to moisture in the atmosphere. Is there no difference in this case?

Response: Thanks for the valuable and professional suggestion. According to the reviewer’s suggestion, we further studied the effect of α-Fe2O3(0001) surface containing H2O on the EPFRs formation. The results showed that EPFRs could be formed on the α-Fe2O3(0001) surface containing H2O through the proton transfer mechanism. This mechanism is different from that of EPFRs formation on α-Fe2O3(0001) surfaces containing ·OH and O3, which follows hydrogen abstraction mechanism. The corresponding revision has been added in the revised manuscript as follows:

  1. lines 296-302 of page 8 in the revised manuscript:

"Furthermore, surfaces may also adsorb species possessing the ability to transfer hydrogen atoms, such as water (H2O), acidic and basic pollutants, thereby influencing the formation of EPFRs. The effect of α-Fe2O3 (0001) surface containing H2O on the EPFRs formation was further studied, as illustrated in Figure S5. The corresponding reaction energy and reaction barrier are -38.0 and 9.5 kJ/mol, respectively, indicating that the EPFRs formation through the proton transfer mechanism is both thermodynamically and kinetically favourable.".

  1. lines 308-311 of page 8 in the revised manuscript:

"In this work, the α-Fe2O3 (0001) surfaces containing ·OH and O3 selected as the interface are beneficial to the EPFRs formation through the hydrogen abstraction mechanism. This provides new insights into the formation of EPFRs in the context of atmospheric composite pollution.".

Figure S5 The energy profile for the reaction of C6H5OH on the α-Fe2O3(0001) surface containing H2O together with the corresponding structures, including initial state (IS), transition state (TS), and final state (FS). Energy is in kJ/mol. Color code: Fe (blue), C (gray), O (red), and H (white).

4) It might be helpful to briefly explain the purpose of each model the authors used.

Response: Thanks for the valuable and professional suggestion. Under complex air pollution, the Fe2O3 surface often adsorbs a large amount of the atmospheric constituents, such as ·OH and O3. Previous research has found that ·OH and O3 quickly adsorb onto the Fe2O3 surfaces, which have great potential to form EPFRs. Thus, the ·OH and O3 are adsorbed on the α-Fe2O3(0001) surface are used as the model for α-Fe2O3(0001) surfaces containing ·OH and O3 in this work. To explore the effect of α-Fe2O3(0001) surfaces containing ·OH and O3 on the formation of EPFRs in the context of complex atmospheric pollution, this model was used. The corresponding revision has been added in the revised manuscript as follows:

  1. lines 71-74 of page 2 in the revised manuscript:

"As is widely recognized, the ·OH and ozone(O3) are prominent oxidants in the atmosphere.24 They exhibit extremely high reactivity in chemical reaction, and thus play a pivotal role in complex air pollution. Furthermore, ·OH and O3 have been observed to quickly be adsorbed on the surface of transition metal oxides and some saline particles.".

  1. lines 82-86 of page 2 in the revised manuscript:

"Therefore, to explore the effect of α-Fe2O3(0001) surfaces containing ·OH and O3 on the formation of EPFRs under conditions of complex atmospheric pollution, models with adsorbed ·OH and O3 on the α-Fe2O3(0001) surface are used to understand the underlying mechanism in the context of complex atmospheric pollution.".
